# A socioeconomic and cost benefit analysis of Tropical Race 4 (TR4) prevention methods among banana producers in Colombia

Thea Ritter [1]*, Diego Álvarez[1], Leslie Estefany Mosquera[1], Edward Martey[1,2]‡, Jonathan Mockshell[1]‡*

1 Performance, Innovation and Strategic Analysis for Impact (PISA4Impact), Applied Economics and Impact Evaluation, International Center for Tropical Agriculture (CIAT), Palmira, Colombia, 2 Socioeconomics Section, CSIR-Savanna Agricultural Research Institute, Nyankpala, Ghana

☯ These authors contributed equally to this work.
‡ EM and JM also contributed equally to this work.
* t.ritter@cgiar.org (TR); j.mockshell@cgiar.org (JM)

**Data Availability Statement:** Not all data can be shared publicly because of concerns of confidentiality. There are restrictions in data availability due to the data containing potentially

## Abstract

The global banana industry faces a significant threat from *Fusarium oxysporum* f. sp. *cubense* Tropical Race 4 (TR4). While prior research has concentrated on TR4's dissemination, reproductive conditions, and resistant banana varieties, this study employs a socioeconomic and cost-benefit analysis to explore the vulnerability of banana producers to TR4 in Colombia. It assesses the financial viability of current monitoring strategies and estimates potential losses in the event of TR4 spreading within the study area. Interviews were conducted with producers and key stakeholders in Colombia's top two banana-producing departments, Antioquia and Magdalena. The findings reveal that farming systems are highly vulnerable to TR4, particularly due to the prevalent use of corms. Producers employ preventive measures such as cement paths, fences, disinfecting stations, and footbaths to counteract TR4's spread. A cost-benefit analysis indicates that the benefits of these prevention methods significantly outweigh the associated costs, with a net present value of implementing prevention strategies per hectare of $95,389 USD and $112,527 USD in Magdalena and Antioquia and a benefit-cost ratio of 3.1 and 4.2, respectively. Considering the substantial impact TR4 could have in Colombia if it becomes more widespread, we recommend widespread adoption of preventive measures, including the construction and utilization of cement paths and disinfectant methods on all banana farms. Additionally, to enhance awareness and early detection, we propose leveraging technology, such as mobile applications (apps) and chat groups, to empower farmers in identifying and preventing the spread of TR4.

## Introduction

The global banana industry faces a significant threat from *Fusarium oxysporum* f. sp. *cubense* (Foc) Tropical Race 4 (TR4; vegetative compatibility group [VCG] 01213/16), a soil-borne fungus affecting the economically important Cavendish variety that blocks the vascular system of

identifying information and the identifying information having the potential to damage respondents' financial opportunities. Some data are available from the authors (contact via Jonathan Mockshell at j.mockshell@cgiar.org) or from the Institutional Review Board (IRB) Chair Juliana Muriel at j.muriel@cgiar.org) for researchers who meet the criteria for access to the data. We will ensure long-term data storage and availability by having our data available on internal servers that are password protected and accessible by all authors of the study in addition to Juliana Muriel.

**Funding:** This work was funded by the Deutsche Gesellschaft für Internationale Zusammenarbeit (GIZ) in the project, "Innovations for the prevention and management of the banana fungal disease Foc TR4 (ALER4TA)", implemented by the Fund for the Promotion in Agriculture (i4Ag) as part of the special initiative Transformation of Agricultural and Food Systems. This research was also financially supported by the CGIAR research initiative on National Policies and Strategies (NPS), which is grateful for the support of CGIAR Trust Fund contributors. The funders had no role in study design, data collection and analysis, decision to publish, or preparation of the manuscript.

**Competing interests:** The authors have declared that no competing interests exist.

banana plants, impeding water and nutrient transport and ultimately leading to plant death. TR4 poses a persistent hazard as it can remain in the soil and plant debris for over 40 years and can also infect and persist in the roots of other hosts, such as close relatives of the banana, weeds, and grasses [1, 2], causing widespread destruction in affected areas. Its spread is mainly the result of human activities and once a farm is contaminated, disease management is difficult and expensive [3]. This fungus not only inflicts substantial economic losses on banana crops but also raises concerns for various other crops, such as tomato, sweet potato, legumes, cucurbits (the gourd family) [4]. Foc's pathogens exhibit diverse and shifting populations as a result of changes in environmental conditions [5]. Race 4 strains of Foc include subtropical race 4 (STR4)–which affects the Cavendish variety under certain physical or abiotic constraints like water logging in subtropical regions–and TR4 [6, 7] which affects the Cavendish variety and a wide array of local cultivars that remain unaffected by Foc Race 1 (R1) and Race 2 (R2) [8]. Remarkably, over 80% of the global banana and plantain production relies on germplasms susceptible to TR4 [9]. There are no commercially available varieties resistant to TR4 [10] and once detected, there is no effective treatment, leading to complete yield loss [11].

TR4 poses a formidable threat to banana cultivation worldwide [12], particularly in Latin America [3]. TR4 was first identified in Asia in the 1990s [13], later spreading to Africa in 2013 [14] and reaching Latin America in Colombia in 2019 [15], Peru in 2021 [16], and Venezuela in 2023 [17]. The arrival of TR4 in Latin America has put the region's banana industry and greater economy at risk [3, 18], particularly in Colombia, Peru, and Venezuela [19–21]. Four of the world's five leading banana exporters are in Latin America: Ecuador, Costa Rica, Guatemala, and Colombia [22]. In Colombia, banana exports were valued at $916.2 million USD in 2020, representing 11.6% of total agricultural exports [23]. Beyond its impact on international trade, the banana sector in Colombia holds significance for food security and employment, providing 293,648 direct and indirect jobs in the country [24]. To address the threat of TR4 to banana production in Colombia, a few key interventions have been implemented: enhancing the capacity of and coordination among stakeholders to manage TR4, reinforcing the use of preventive measures through surveillance, and containing existing cases of TR4 [18].

Despite the importance of bananas in Colombia and the recent arrival of TR4 there, there are no studies on the socioeconomic consequences of TR4 for value chain actors on the effectiveness of interventions in a Latin American country. Important aspects such as the disease's effects on food security and the livelihoods of value chain actors are absent. Instead, the scant literature focuses on examining epidemiology, pathogen spread factors, and disease management [9, 20, 21, 25–29], the efficacy of control methods [30, 31], overall costs of TR4 at the farm-level [32] and country-level in Asia [33], and global consequences of TR4 and the delay in adopting resistant varieties [34]. A few studies focus on evaluating the impact of strategies to mitigate the effects of TR4 and other banana diseases on poverty reduction and avoided production losses [9, 35, 36]. These studies use methodologies such as cost benefit analysis (CBA), net present value (NPV), internal rate of return (IRR), and partial equilibrium economic surplus models to compare strategies related to the adoption of different scenarios and spread rates. For instance, Scheerer et al. [35] evaluate the economic viability of implementing strategies to mitigate the impact of Fusarium (R1, R2, STR4, and TR4) in 29 countries in Asia, Africa, and Latin America. Results indicate that, assuming a 50% spread rate, production losses over a 25-year horizon could reach nearly 1.7 million hectares [35]. Applying a partial equilibrium economic surplus model and CBA, Staver et al. [9] examine the impact of four mitigation options against TR4, finding that all options generate positive NPV and that the IRR is above the standard 10%. The above studies show the immense potential impacts of TR4 and the benefits of proactively detecting and preventing its spread.

This study aims to fill the existing gaps in the literature by evaluating the socioeconomic effects of TR4 and the financial viability of interventions used for disease management of TR4 in Colombia based on data collected from banana producers and other stakeholders. Through a CBA and socioeconomic analysis, this study aims to answer the following research questions: (1) How vulnerable are banana producers and current banana farming systems to TR4? (2) What disease management strategies are currently implemented on banana farms? (3) What is the financial viability of these strategies in terms of their NPV and benefit-cost ratio (BCR)? (4) What are the estimated economic losses if TR4 were to spread? The findings will provide actionable insights into the potential of technology, policies, and institutional responses for preventing and managing the spread of TR4, thereby helping policymakers, extension agents, scientists, development organizations, and other stakeholders develop and target policies and initiatives to help combat the spread of TR4.

## Materials and methods

### Study area

The departments of Antioquia and Magdalena in Colombia were selected due to several strategic considerations (Fig 1). These departments stand out as the primary hubs for banana

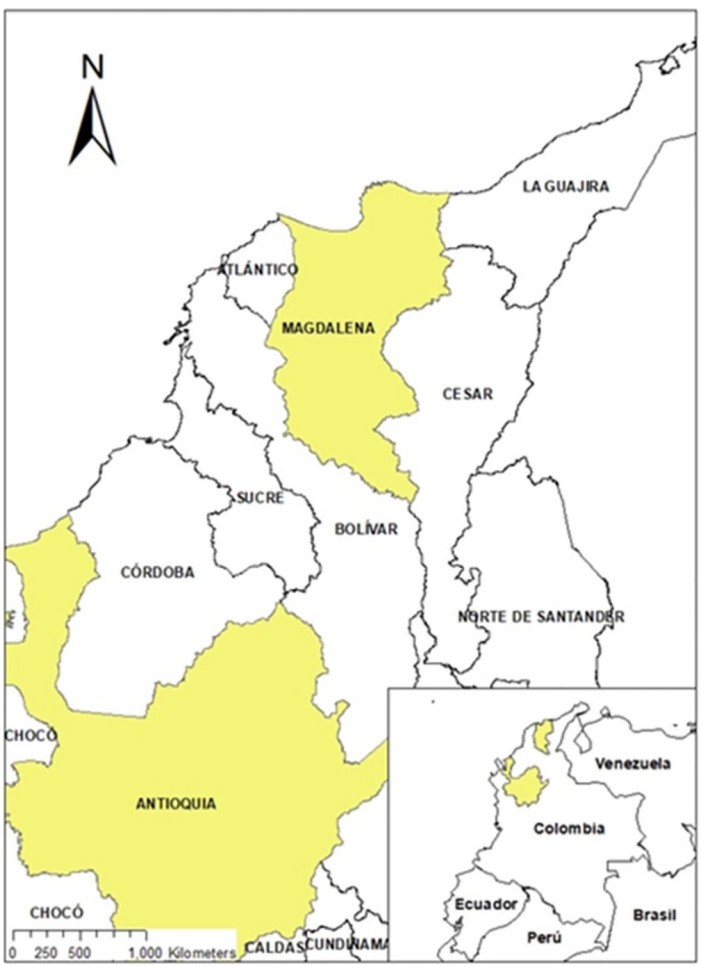

**Fig 1. Map of departments of study in Colombia.** Source: Own elaboration based on survey.

production in Colombia. Second, they exhibit distinct farming systems: Antioquia relies on large plantations with hired labor, while Magdalena predominantly features smaller farms relying on family labor. This deliberate choice enables a comparative analysis of disease management practices across different production systems. Third, Antioquia and Magdalena produce bananas for different markets: Antioquia is the only department in Colombia that produces for both export and domestic markets, whereas Magdalena produces only for international markets. The two major varieties sold in Colombia are Cavendish for export and Banano Criollo for domestic markets. Lastly, the inclusion of Magdalena is motivated by the presence of three confirmed TR4 cases [37], providing an opportunity to investigate areas both with and without TR4.

## Data collection and sources

Prior to initiating data collection, a workshop under the project, "Innovations for the prevention and management of the banana fungal disease Foc TR4 (ALER4TA)" project was conducted in Santa Marta, Magdalena, from October 24th–25th, 2022 to ensure stakeholders were well-informed about the project's objectives and the nature of the data to be gathered. The workshop also served to identify existing TR4 prevention actions undertaken by industry participants in Colombia, offering valuable insights that shaped the subsequent data collection process.

Both quantitative and qualitative data were gathered through a comprehensive approach involving a producer survey and semi-structured interviews (SSIs). Ethics approval was obtained for this study from the Institutional Review Board of the Alliance of Bioversity International and International Center for Tropical Agriculture. There were no permits required for access to the study's field sites. Before beginning the interviews, respondents provided verbal consent for their participation and were informed that they could withdraw their consent and discontinue participation at any time. Enumerators recorded each respondent's verbal consent on the survey form.

The producer survey included 191 smallholder farmers and large plantation owners/managers (see Table 1). The number of surveys conducted in each municipality varies widely given the different production systems: A higher number of interviews were conducted in Magdalena (176), predominantly due to the prevalence of smaller-sized farms in comparison to Antioquia (15). The SSIs provided valuable insights into the diverse effects of TR4 that may not have been captured by quantitative data alone. Before collecting data, we met with government officials and agricultural experts in Colombia's banana sector to determine which actors to interview. To understand the varied effects of TR4 throughout the value chain, we conducted 28 SSIs with a range of stakeholders, including extension agents, banana laborers, researchers,

**Table 1. Sampled actors interviewed.**

| Type of actor | Number of interviews |
|---|---|
| Producers | 191 |
| Extension agents, service providers, and multiplicators | 9 |
| Banana laborers | 10 |
| Researchers | 2 |
| Presidents/Vice Presidents of producer organizations | 5 |
| Government officials | 2 |
| Total | 219 |

Source: Own elaboration based on semi-structured interviews.

representatives from producer organizations, and government officials (Table 1). Interviews were conducted between February 21st–July 6th, 2023.

Sampling support was restricted due to privacy concerns. Because cooperatives and producer associations were unwilling to provide us with a list of their members, we asked cooperatives to allow us to utilize their facilities to conduct surveys with producers who were visiting the cooperative offices to receive payment or training. All banana producers visiting four cooperatives were asked to participate in the survey during the day the enumerator team was present in each cooperative. About 60% of producers approached in cooperatives declined to participate in the survey. To try to find more producers to interview, including those not in cooperatives, cooperative leaders guided the enumerator team to villages with banana producers. All banana producers located during 11 days of field work in villages were asked to participate in the survey with about half saying they were not willing. Data collection took longer than planned to complete for a few reasons, namely unforeseen challenges contacting actors to interview, coordinating the schedules of associations and cooperatives to interview members, and trying to interview respondents with TR4, as explained below.

Initially, we set out to interview producers both with and without TR4 on their banana farms. Despite the confirmation of TR4 on large farms in La Guajira and Magdalena, attempts to interview producers or managers from these farms were unsuccessful. Despite assuring anonymity and employing various outreach methods for over four months, producers–especially those who had encountered TR4 –were unwilling to participate or provide information due to concerns about potential harm to their economic activities. Producers expressed fear that revealing their TR4 situation might hinder their chances of obtaining necessary farm loans. Consequently, only producers without TR4 agreed to be interviewed and thus this study was not able to analyze the costs associated with eradication of infected plantations. Because the surveyed producers had training on TR4 and actively apply prevention measures to deter the entry of TR4 on their farms, the data reflect real, proactive efforts to mitigate potential risks. In addition to data on TR4 training and prevention measures, the survey collected data on income from banana production, income from banana labor, the banana production system, participation in producer organizations, banana consumption at home, and food security. Food security was assessed using the Food Insecurity Experience Score [38], which is based on eight questions about respondents' experiences over the last 12 months, such as skipping meals, and categorizes responses into levels of food insecurity.

Data from surveyed producers was used to estimate the construction costs of disinfection stations and cement paths. These costs included initial construction expenses, annual depreciation, regular maintenance, and necessary operational inputs (such as disinfectants, boots), and supervision. The costs of cement paths were calculated based on an average of 1.3 meters of cement paths per hectare, which is derived from information provided by producers. In addition to primary data, we also utilized secondary data to estimate costs and simulate the annual economic losses due to TR4 across four dispersion scenarios at the department and producer-level. Average operational costs per hectare for banana cultivation in each department were estimated based on publicly available data [24]. We were unable to obtain cost data from producers because they were unwilling to share this information. The simulation utilized data on hectares planted with bananas and departmental production in 2021 from the Municipal Agricultural Assessments conducted by the Colombian Ministry of Agriculture and the average sales price from surveyed producers. Estimations of the losses caused by TR4 dispersion scenarios utilize data on hectares planted with bananas and production in 2021 from the Municipal Agricultural Assessments conducted by the Colombian Ministry of Agriculture and the number of potential jobs lost were estimated using the average of 0.8 direct jobs and 2.8 indirect jobs per hectare of banana [39]. The value of direct employment was used by

calculating the average daily wage from the producer survey and the value of indirect employment was calculated with the legal minimum wage in Colombia in 2023 [40]. Export losses were calculated based on average sales prices from the producer survey. Colombia has regulations governing the use of TR4 contaminated land: Subject matter experts and stakeholders stated in various workshops and meetings that growing bananas or any other crops on land infected by TR4 is not permitted, and that if producers want to do anything on such land, they need special permission by ICA [41]. As a result, losses in land value were calculated based on the number of affected hectares multiplied by the annual land rental price from the producer survey. Results at the producer-level were based on the average producer in each department based on data from the producer survey.

Lastly, while CBA can help stakeholders make rational decisions in complex situations, quantifying intangible benefits (such as by utilizing an index or monetizing intangible benefits) is difficult and subjective [42, 43]. Previous research has found that there are intangible costs of Fusarium wilt, such as fear, agricultural equity, social exclusion, loss of markets, insecurity, and isolation [32, 44]. Therefore, to capture intangible aspects of TR4 that the CBA is unable to quantify [45], producers were asked about the impact of TR4 on the economy, their jobs, food security, and the environment.

## Data analysis process

Data from the producer survey and SSIs were analyzed separately to examine the effects of TR4 on various indicators, such as food security, income, and the detection and management of TR4. From the SSIs, open-ended questions and multiple-choice questions were tabulated in Excel.

This study uses a CBA approach to evaluate the potential benefits of TR4 prevention strategies. CBA is an analytical tool that balances the costs and benefits of a project to evaluate its financial profitability [46]. There are several factors necessary for the CBA calculation. The first is the analysis horizon, which is the time horizon or useful life of the project. Consulting the Colombian National Tax Statute Article 137, we selected a 40-year analysis horizon as the foundational useful life for the disinfection stations and path cement. An annual depreciation rate of 2.22% and 2.5% was applied to disinfection stations and cement paths, respectively. The second factor is the discount rate, which is used to discount future payments to their present value, thereby accounting for the time value of money. We chose a 12% social discount rate [47], which reflects society's time preference and allows for the comparison of cost and benefit streams on an equivalent basis in the present. The third factor is the investment cost, which is the initial cost of a prevention strategy. The fourth factor is operation and maintenance costs, which are all the costs necessary for the development and implementation of a strategy, such as fixed costs, input costs, repairs, and maintenance. All values are adjusted for inflation.

To calculate the NPV and BCR, we combined data from several sources. The following steps were undertaken in the CBA:

**Step 1:** Identify the specific activities of each of the TR4 prevention strategies through a workshop with subject matter experts and key stakeholders.

**Step 2:** Collect data on TR4 prevention strategies applied in the field and their cost structure. As producers were reluctant to disclose cost information, we relied on data from various sources [48, 49] and confirmed the results with an independent expert specializing in banana production in Colombia.

**Step 3:** Clean and analyze the data to ensure uniform units for cost and income information, specifically on a per-hectare basis. This standardization clarifies the financial viability at the

level of individual plots. The average yield per hectare is multiplied by the average selling price of bananas based on data obtained from the producer survey. This provides an estimate of the income that would be preserved due to TR4 prevention strategies.

**Step 4:** Identify the benefits linked to TR4 prevention strategies, considering benefits as the monetary losses averted. This entails estimating the losses that would have occurred on the farm if banana plants had contracted TR4. Following the approach used in Cook et al. [26], these averted losses are termed exclusion benefits.

**Step 5:** Calculate net cash flows by subtracting total income from total costs.

**Step 6:** Calculate NPV, a financial metric which is the difference between the present value of cash inflows and outflows associated with an investment. NPV helps assess the profitability or financial viability of an investment by determining whether it generates more value than it costs when considering the time value of money. A positive NPV indicates that the investment is likely to be financially worthwhile, whereas a negative NPV suggests otherwise. To calculate NPV, one needs to consider the time horizon, net cash flows, and the discount rate, all of which are incorporated into the following NPV formula:

$$\sum_{t=0}^{n} \frac{Net\ cash\ flow_t}{(1+\imath)^t} = NPV \tag{1}$$

where $n$ is the total number of time periods, $i$ is the discount rate, and $t$ is the time period.

**Step 7:** Adjust the income (benefit) and cost of each period to the present using the discount rate selected in the following formula:

$$\frac{Income_t}{(1+\imath)^t} = Current\ benefit \tag{2}$$

$$\frac{cost_t}{(1+\imath)^t} = Current\ cost \tag{3}$$

**Step 8:** Estimate the BCR, which is obtained by dividing the sum of current benefits by the sum of current costs. A ratio greater than one indicates that the project will generate more benefits than costs, therefore making it financially profitable. The BCR equation is:

$$\sum_{t=0}^{n} \frac{Current\ Benefit_t}{Current\ cost_t} = BCR \tag{4}$$

By utilizing key financial metrics, including BCR and NPV, the analysis aims to offer a thorough insight into the economic feasibility of TR4 prevention strategies implemented by banana producers. Economic losses associated with TR4 are estimated across four dispersion scenarios (25%, 50%, 75%, and 100%), guided by the work of Scheerer et al. [35, 36].

## Results

### Background information on surveyed producers and farming systems

**Gender, education, and primary occupation of producers.** Gaining insights into the demographic characteristics of respondents is crucial for understanding the diversity of

experiences and perspectives. A minority of respondents (16.2%) are women, with a slightly higher proportion in Antioquia (20%) compared to Magdalena (15.9%). All respondents in Magdalena indicated that they are able to read and write, while in Antioquia, 3.4% are unable to do so. Educational attainment in Magdalena is diverse: approximately 35.1% have a high school education, 19.3% completed technical studies, 18.1% pursued university studies, and 5.1% had not completed any formal education. In Antioquia, the majority of producers (80%) have pursued university studies, 13% completed technical studies, and 7% finished high school. Regarding their primary occupation, most producers in Magdalena are owner-operators (73.1%), while 11.4% manage other farms. In Antioquia, 46.7% of respondents work on their own farm as their primary occupation, 13.3% work as managers on other banana farms, and 40% are involved in various other farming activities, such as managerial, agronomist, and administrative roles.

**Food security.** The results indicate that producers in Antioquia have adequate access to food, as all surveyed producers did not indicate that they face food insecurity. However, in Magdalena, the scenario is different, with 20.45% of producers experiencing mild food insecurity, 18.75% facing moderate insecurity, and 22.73% suffering from severe food insecurity (see Fig 2).

**Farm size and systems.** Farms are significantly larger in Antioquia than in Magdalena. The difference in the average farm size between Antioquia (134 ha) and Magdalena (12 ha) is statistically significant at the 1% level (see Table A in S1 Table). In Magdalena, 59% of respondents are small-scale producers (farms less than 3 ha), 29% medium-scale (farms 3 to 20 ha), and 12% large-scale (farms greater than 20 ha). All 15 respondents in Antioquia are large-scale producers. Conventional production systems dominate, with only 4% employing organic production methods. Marketing companies play an important role in pest and disease control, as they carry out aerial spraying against Black Sigatoka.

While producers in Antioquia rely on hired labor, those in Magdalena rely on a mix of family and hired labor. Very few producers in Magdalena work alone (7.9%). Most rely on hired labor (61.9%). A large share of these producers use only family labor (44.3%) and 21% use a mix of family and hired labor. About one in six (17.8%) family laborers are women.

**Seeds.** Understanding the source of seeds is crucial, especially considering that TR4 can be transmitted through infected soil. In Magdalena, the majority of producers (65%) primarily use corms as their source of banana seeds, followed by seedlings (35%). The majority independently source seeds, often taking them from other plots (79%). Some producers purchase certified seeds (20%) or receive seeds as gifts (1%). In contrast, all producers in Antioquia plant seeds from their own plots. Additionally, 6.7% of Antioquia's producers choose seedlings, while 93.3% prefer corms as their primary seed source. The differences in the use of seedlings

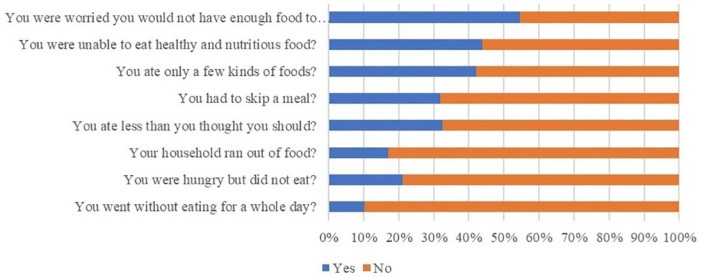

**Fig 2. Food security of banana producers in Magdalena (N = 176).** Source: Own elaboration based on the Food Insecurity Experience Score in the producer survey. Antioquia is not shown since all surveyed producers responded "no" to all questions.

and corms between producers in Magdalena and Antioquia are statistically significant at the 5% level (see Table A in S1 Table). Just over half of producers in both departments were aware of ICA-certified establishments for the sale of banana seeds.

## Markets

Banana production in the study area primarily targets export markets with all producers engaging in the sale of bananas in international markets. A large majority of the harvest is destined for export, with producers from the two departments selling on average 95% of the harvest to export markets The remaining 5% is either sold domestically or utilized for household consumption by the producers themselves. Bananas that fail to meet export standards, referred to as discarded bananas, are sold in local markets at reduced prices. Producers can receive higher prices for certified bananas. Rainforest Alliance, Global G.A.P., and Fairtrade are the most common certifications. All 15 producers in Antioquia have at least one certification, whereas in Magdalena, 79.89% of producers are certified. Income data from nine producers in Antioquia and 144 in Magdalena indicate a significant disparity in average gross income from banana sales in 2022. Producers in Antioquia report an average income of around $2 million USD, while those in Magdalena earn an average of $143,000 USD, highlighting distinct socio-economic differences between the two regions.

**Membership in associations/cooperatives.** In Colombia, producer associations and cooperatives play a crucial role in organizing and supporting banana producers in selling their products to international markets. A high proportion of the sampled producers are members of such associations, accounting for 80% in Antioquia and 85.23% in Magdalena. Membership in cooperatives offers several advantages, including the ability to secure better prices, facilitate transactions, access technical assistance, purchase inputs at reduced costs, and negotiate higher prices. The SSIs revealed that cooperatives are also instrumental in TR4 prevention efforts. They provide TR4 mitigation kits, including footbaths, back pumps, pressure washers, full-body clothing, and five pairs of boots, valued $688 USD. Financial support for these initiatives is derived from a partnership between banana producer associations and the Colombian government.

## Awareness, training, and monitoring strategies for TR4

No significant differences exist between the two departments in terms of overall awareness of and access to training on TR4. In Antioquia, all 15 producers were aware of TR4, with 14 having received training on the topic. These producers were trained an average of eight times on TR4 in 2022. Conversely, in Magdalena, 174 out of 176 banana producers were aware of TR4 and 161 participated in an average of 5.8 training sessions on TR4 in 2022. The providers of these crucial trainings were producer associations (59.6%), the government's Colombian Agricultural Institute (ICA) (52.8%), and marketing companies (20.5%). Extension agents trained a higher proportion of producers in Antioquia, whereas producer associations and cooperatives were the main providers of training in Magdalena.

There are some statistically significant differences regarding various monitoring strategies between the two departments (see Table A in S1 Table). In Antioquia, all producers have implemented monitoring measures. While 27% rely on visits from extension agents to help monitor their banana plant, only 13.3% use mobile apps during the monitoring process. In Magdalena, 89% of respondents have implemented monitoring strategies, with 58.6% conducting symptom monitoring through direct involvement or with the assistance of farm workers, 39% relying on extension agents, and 5% utilizing mobile applications. One producer in Antioquia and another in Magdalena use drones for symptom monitoring. Cooperatives and

marketing companies play a role in surveillance by supporting producers in the supervision of banana plantations, as these institutions send technicians to monitor the phytosanitary status of farms.

The frequency of monitoring activities varies based on the strategies employed. When symptom monitoring is conducted by producers or farm workers, it occurs daily as part of the farm's routine activities, yet monitoring by extension agents or drones typically occurs monthly. The majority (79%) of the costs for monitoring by extension agents is financed by cooperatives, associations, government institutions, or marketing companies. All producers in Antioquia and 98% in Magdalena have implemented prevention strategies, which are described below.

**Disinfection stations and foot baths.** Disinfection stations play a crucial role in controlling the spread of TR4. Typically located at the farm's entrance, these stations consist of two zones separated by a cement bench. The first zone serves as the "dirty zone" where individuals leave their regular shoes, while the second zone is the "clean zone." To leave the disinfection station, one must pass through a footbath with a disinfecting solution. The implementation of these stations ensures that visitors and farm workers do not inadvertently carry the fungus on farms. Remarkably, all respondents in Antioquia have built disinfection stations, whereas 78% have in Magdalena. The difference is statistically significant at the 5% level (see Table A in S1 Table). Many disinfection stations on small-scale farms were purchased by cooperatives. In addition, some inputs, such as quaternary ammonium and boots, were supplied by associations and ICA. Foot baths to disinfect shoes before and after entering farms are another strategy. Foot baths contain a mix of ammonium quaternary and water. They are strategically placed in various zones of the farm, especially at different entrances. This additional layer of protection helps eliminate potential soil contamination. In Antioquia, 73% of respondents have foot baths, which is slightly higher than in Magdalena (69%), yet the difference is not statistically significant (see Table A in S1 Table).

**Cement paths and fences.** Other strategies are aimed at building physical barriers. One of these is cement paths, which are delimited paths between the entrance to the farm, the place of production, and administrative areas, to help minimize soil-to-plant contact, thereby reducing the risk of disease transmission. Cement paths minimize direct contact between soil and footwear, tools, vehicles, and animals. They prevent workers or visitors from direct soil contact within banana farms, thus avoiding soil movement and preventing the unintentional spread of contaminated soil on footwear or tires. In Magdalena, 5% of respondents have built cement paths while 33% have in Antioquia: This difference is statistically significant at the 1% level (see Table A in S1 Table). Fences are very helpful for producers since they prevent animals and people from walking across multiple plantations, which could spread TR4.

**Technology access and its use to manage TR4.** Access to information and technology varies widely in the study area. Although most producers in Magdalena own a smartphone (80%), there is a connectivity gap since 24% lack internet access. Among those with internet access, 50% use a combination of mobile data and Wi-Fi, 28% use only mobile data, and 22% use only Wi-Fi. Internet access in Magdalena is primarily through smartphones (99.3%), followed by personal computers (27.8%), public computers (1.5%), and tablets (0.7%). In contrast to Magdalena, all respondents in Antioquia have smartphones and internet access: 60% access the internet using both mobile data and Wi-Fi and 40% rely solely on Wi-Fi. Producers there access the internet through their smartphones and 73.3% also use personal computers.

As shown above, mobile phones, especially smartphones, are the primary means of internet access. Producers use the internet to access various types of agricultural management information (66%), input information (53.5%), and to seek guidance on pests and diseases (42.4%). Fusarium is the most searched disease (76.2% of producers searched for it in the past year),

followed by Black Sigatoka (59.5%). According to extension agents and service providers, technology is useful for TR4 prevention when it is used to provide "timely and accurate responses" and "accelerate the protocols for early symptoms". Further, extension agents believe that technology will be useful in the future by providing satellite spectrums with accurate location and by shortening detection times for precise information about the location of outbreaks. In terms of TR4 management, extension agents agree that technology is key for obtaining "precision and follow-up information", "creating new varieties resistant to the fungus", and "improving the quality of the information". They also mentioned that technology will be more useful in the future by "shortening time for early detection" via "better identification of symptoms" with "apps useful to detect, inform, and locate." Other types of actors interviewed had similar opinions. Cooperative managers and government officials viewed the importance of technology for TR4 management in terms of obtaining and sharing information, while researchers mentioned developing new banana varieties and improving disease detection times.

The use of mobile phone apps for crop management is much lower than the use of chat groups. A relatively small share of the total sample (16%) utilize apps for crop management, with a much larger share utilizing such apps in Antioquia (60%) compared to Magdalena (11.4%). Some commonly used apps in Magdalena include Avanza, Banasoft, Excel, Farmers Edge, Field Climate, Georeferencing and GPS, Planty, Silver Tracker, and WhatsApp. In Antioquia the apps Xioma and Tropimovil are the most common. These apps primarily serve the purpose of obtaining information about agricultural management, market trends for bananas, and inputs. Banana producers in both departments actively engage in chat groups to exchange valuable information about bananas. In Antioquia, 60% of banana producers actively participate in such chat groups, with all of them using WhatsApp and 22% using Facebook groups. Similarly, in Magdalena, 48.3% of respondents participate in similar groups, with all using WhatsApp and just 2.3% using Facebook or 1.2% Telegram. These results show that banana producers connect with one another and seek advice through their mobile phones.

## Cost benefit analysis (CBA) of implementing TR4 prevention measures

**Costs estimation of TR4 disinfection stations, footbaths, and cement paths.** Table 2 provides a comprehensive overview of the costs per hectare associated with the establishment of disinfection stations and cement paths. Given that Antioquia boasts large-scale producers

**Table 2. Costs of disinfection stations and cement paths (in USD per hectare).**

|  | Magdalena (N = 73) | Antioquia (N = 6) |
|---|---|---|
| **Disinfection stations** | | |
| Construction (setup costs in year 0) | $396 | $258 |
| Depreciation (annual) | $9 | $5.70 |
| Maintenance (annual) | $119 | $128.60 |
| Supervision (annual) | $1,863 | $1,647 |
| Water (annual) | $0.03 | $2.21 |
| Quaternary ammonium cost (annual) | $59.61 | $673 |
| Boots (annual) | $35.60 | $39.30 |
| **Cement paths** | | |
| Construction (setup costs in year 0) | $98 | $28.50 |
| Depreciation (annual) | $2 | $0.70 |
| Maintenance (annual) | $4 | $1.10 |

Source: Own elaboration based on producer survey.

**Table 3. Average yield, price, and economic benefits by department (N = 183).**

|  | Magdalena (N = 169) | Antioquia (N = 11) |
|---|---|---|
| Yield (tons/ha) | 38.66 | 46.24 |
| Price per ton of bananas | $395.90 | $355.20 |
| Total benefits (avoided losses per hectare) | $15,305.49 | $16,424.45 |

Source: Own elaboration from producer survey.

and Magdalena predominantly consists of small and medium-sized producers, the disparity in costs can be explained by the fact that the costs per hectare tend to decrease with farm size. For example, maintenance costs are lower per hectare in Antioquia because they can buy inputs in bulk.

**Benefits as avoided losses: Exclusion benefits.** Table 3 presents information on average banana yields, the price of bananas per ton, and the substantial financial benefits arising from the implementation of prevention strategies in Magdalena and Antioquia. In Magdalena, the average yield is 38.66 tons of bananas per hectare and the average price is $395.90 USD/ton, while in Antioquia, the average yield is higher (46.24 tons/hectare) and the average price is lower ($355.20 USD/ton). However, the differences between the average yield and prices for each region are not statistically significant (see Table A in S1 Table).

**Net present value and benefit-cost ratio (BCR).** The cost of production per hectare in Magdalena is estimated to be $2,854 (USD/ha), while it is $3,230 (USD/ha) in Antioquia. Production costs in Antioquia are elevated due to the increased expenses associated with aerial spraying for Black Sigatoka, a banana plant disease more prevalent in Antioquia than in Magdalena, attributable to higher rainfall in the former. Table 4 summarizes the financial performance metrics of investing in prevention strategies, namely disinfection stations and cement paths, in Magdalena and Antioquia. As shown in Table 4, in Magdalena, the NPV per hectare is $95,389 USD, while Antioquia records a higher NPV of $112,527 USD. The very high NPV values indicate that the benefits derived from implementing these prevention strategies outweigh the initial costs, and that the investments are expected to generate favorable returns over time. This economic perspective underscores the importance of adopting and sustaining prevention strategies to safeguard the future of banana cultivation.

The BCR in both departments is greater than 1 (see Table 4), indicating that the investment will generate more benefits than it costs. In Magdalena, the BCR is 3.1. This implies that each unit of investment in both banana production and prevention measures (disinfection station, disinfection of footwear and cement paths) is expected to generate 3.1 units of benefits; i.e., the benefits are more than three times greater than the costs. Similarly, in Antioquia, the financial outlook is also very favorable, with an BCR of 4.2. These robust returns underscore the substantial financial advantages prevention strategies offer, regardless of farm size.

**Table 4. Net present value and benefit-cost ratio by department (N = 183).**

|  | Magdalena (N = 169) | Antioquia (N = 11) |
|---|---|---|
| Net Present Value (NPV) | $95,389 | $112,527 |
| Benefit-cost ratio (BCR) | 3.1 | 4.2 |

Source: Own elaboration based on producer survey.

## Economic losses from different dispersion scenarios

We simulated the annual economic losses due to TR4 across four hypothetical dispersion scenarios. Doing so allows a more precise and quantitative view of the possible economic consequences of TR4 on the local banana industry. Table 5 shows the potential annual consequences in terms of banana production, employment, exports, and land value in Magdalena and Antioquia based on different TR4 dispersal rates (25%, 50%, 75%, and 100%). The findings indicate that even under moderate dispersion, significant losses are evident. In Magdalena, moderate dispersal (25%) already entails substantial losses, with thousands of hectares eradicated and significant tonnage of produce lost. However, as the pace of dispersal accelerates, the consequences become immense. In the full dispersal scenario, over 18,000 hectares of crops and 700,000 tons of annual production would be lost, which would mean annual economic losses from banana exports could reach $250 million USD, constituting 33.7% of the agricultural gross regional product of Magdalena [50]. In addition, over 14,000 direct jobs (such as agricultural workers, machinery operators, packers, and transport drivers) and 50,000 indirect jobs (such as suppliers of agricultural inputs, logistics and transportation services, financial services, and roles in marketing and sales) would be lost, representing a total of $238.4 million USD in salaries. If soils in Magdalena were to become infected, estimated losses in land value would approach $25 million USD. TR4 dispersion in Antioquia would also generate vast economic consequences. In the scenario of moderate dispersion, nearly 10,000 hectares of land would be affected from TR4, resulting in the loss of over 325,000 tons of bananas annually. As TR4 dispersal intensifies to 50%, 75%, and complete dispersal (100%), the consequences worsen, reaching nearly 40,000 hectares, representing a loss in land value of approximately $54.7 million USD and over 1.3 million tons of annual banana production lost. The potential economic

**Table 5. Potential annual losses from the spread of TR4 at the department level.**

|  | Affected area (ha) | Potential annual lost production (tons) | Potential employment losses | | | | Export losses per year (M USD) | Loss in land value (M USD) |
|---|---|---|---|---|---|---|---|---|
|  |  |  | Direct employment | | Indirect employment | | | |
|  |  |  | People (N) | Economic losses (M USD) | People (N) | Economic losses (M USD) | | |
| **Magdalena** |  |  |  |  |  |  |  |  |
| Moderate dispersion (25%) | 4,512 | 178,655 | 3,609 | $ 13.2 | 12,634 | $ 46.4 | $ 62.5 | $ 6.2 |
| Accelerated dispersion (50%) | 9,024 | 357,310 | 7,219 | $ 26.5 | 25,267 | $ 92.7 | $ 125.1 | $ 12.5 |
| Rapid dispersion (75%) | 13,535 | 535,965 | 10,828 | $ 39.7 | 37,898 | $ 139.1 | $ 187.6 | $ 18.7 |
| Total dispersion (100%) | 18,047 | 714,621 | 14,438 | $ 52.9 | 50,532 | $ 185.4 | $ 250.1 | $ 24.9 |
| **Antioquia** |  |  |  |  |  |  |  |  |
| Moderate dispersion (25%) | 9,874 | 325,935 | 7,899 | $ 41.4 | 27,647 | $ 144.9 | $ 114.1 | $ 13.7 |
| Accelerated dispersion (50%) | 19,748 | 651,870 | 15,799 | $ 82.9 | 55,294 | $ 289.9 | $ 228.2 | $ 27.3 |
| Rapid dispersion (75%) | 29,622 | 977,806 | 23,698 | $124.3 | 82,294 | $ 431.6 | $ 342.2 | $ 40.9 |
| Total dispersion (100%) | 39,496 | 1,303,741 | 31,597 | $165.7 | 110,589 | $ 579.9 | $ 456.3 | $ 54.6 |

Source: Based on data from the Ministry of Agriculture [24], UGRA [39], Banrep [40], and the producer survey.

Notes: N is number and M is million.

**Table 6. Potential annual losses from the spread of TR4 at the producer-level.**

|  | Affected area (ha) | Banana production (tons) | Export income (USD) | Domestic sales income (USD) | Land value (USD) |
|---|---|---|---|---|---|
| **Magdalena** | | | | | |
| Moderate dispersion (25%) | 3 | 121 | $ 115,226 | $ 2,079 | $ 4,218 |
| Accelerated dispersion (50%) | 6 | 243 | $ 230,452 | $ 4,158 | $ 8,436 |
| Rapid dispersion (75%) | 9 | 364 | $ 345,677 | $ 6,237 | $ 12,654 |
| Total dispersion (100%) | 12 | 485 | $ 460,903 | $ 8,315 | $ 16,872 |
| **Antioquia** | | | | | |
| Moderate dispersion (25%) | 34 | 1,527 | $ 504,688 | $ 32,073 | $ 46,537 |
| Accelerated dispersion (50%) | 67 | 3,054 | $ 1,009,377 | $ 64,147 | $ 93,075 |
| Rapid dispersion (75%) | 101 | 4,580 | $ 1,514,065 | $ 96,220 | $ 139,612 |
| Total dispersion (100%) | 135 | 6,107 | $ 2,018,753 | $ 128,293 | $ 186,149 |

Source: Own elaboration based on data from the producer survey (N = 191).

Notes: Average areas, production, and export value per year per hectare are presented for the average farmer in Magdalena and Antioquia.

losses from banana exports could amount to $456 million USD annually in the worst-case scenario, posing a severe threat to local businesses and impeding overall economic growth. Such a loss represents 15.4% of the agricultural domestic product of Antioquia [50].

Table 6 provides estimates of the annual economic impact of TR4 on the average banana producer in each department. The results show that the impact of TR4 is expected to vary in Magdalena and Antioquia. Given the larger farm size and higher yields in Antioquia, the impacts on individual producers are higher. For example, estimated losses in banana production for the average producer in the total dispersion scenario are 485 tons and 6,107 tons in Magdalena and Antioquia, respectively.

## Intangible aspects of TR4

To supplement the quantitative data on the costs and benefits of TR4 prevention strategies, producers were asked to respond to questions related to TR4. Their responses highlight some of the intangible costs of TR4. For example, when asked to describe the effects on the economy if TR4 were to spread in their area, commonly used words include exports, income, economy, loss, finished, reduced, food, and nutrition. Commonly used words to describe how TR4 would affect their jobs include unemployment, social, families, livelihood, and layoffs. Words to describe the impact of TR4 on food security include people, income, work, money, unemployment, catastrophe, serious, fear, debts, crisis, and families. Lastly, words to describe the implications of TR4 on the environment include land, chemicals, crops, soil, pollution, water, and contamination. These results highlight the widespread impact further spread of TR4 would have in banana producing areas and provide insights into the intangible benefits of TR4 prevention.

## Discussion

The findings from this study underscore that interventions supporting TR4 prevention strategies can yield significant positive impacts, potentially preventing substantial economic losses. By understanding the current state of banana farms, current monitoring strategies, the financial viability of prevention methods, and the economic impact if TR4 were to spread, stakeholders can better design strategies to mitigate the effects of TR4. In this section, we discuss the main results before addressing the study's limitations.

## Vulnerability of banana farming systems and households to TR4

TR4 poses major economic and food security risks for Latin America, especially in countries where it has already been detected [20, 51, 52]. Its impact is widespread and depends on containment and producers' financial capacity to cope with the consequences of the disease, which include increased production costs and decreased production. In the case of Peru and Colombia, the spread of TR4 would lead to export losses [18]. In addition, TR4 has also been detected in Venezuela [17], which raises concerns about the spread of the disease endangering banana production for export and domestic markets, which would affect local economies and food security [3].

This study shows that despite variations in farm size and labor across both departments, banana farms and associated value chain actors in Antioquia and Magdalena are highly vulnerable to the spread of TR4. The rapid and devastating impact of TR4 on infested soils poses a significant threat [53, 54]. Common agricultural practices identified in this study that highlight the vulnerability to TR4 include the use of corms from existing plots for replanting or establishing new plantations. It is crucial to acknowledge that corms could carry pathogens, including TR4: If the corms used as seeds are infected, they have the potential to introduce these pathogens to new areas or facilitate their transmission within an existing plantation [53]. This scenario can lead to the swift dissemination of TR4.

Given that farming is the primary occupation for most producers, if TR4 were to spread to their farms, they would need to find off-farm work to support their families since they are not allowed to grow crops on land infected with TR4. Producers may already need to find more suitable crops to grow, such as paddy rice, sugarcane, tapioca, onion, and pineapple which are non-host crops for TR4 and reduce the inoculum level of the pathogen [2]. Or they may need to seek off-farm employment opportunities to help hedge the risk of TR4 on their farms [55].

## Current monitoring strategies and their financial viability

Proactive prevention strategies demonstrate the dedication and awareness of banana producers in Antioquia and Magdalena in safeguarding the banana industry against TR4. Proactive measures adopted by farmers include surveillance of banana plants, building disinfection stations and cement paths, and disinfecting tools, machinery, and equipment. As shown in Table A in S1 Table, there are statistically significant differences in the adoption of some measures when comparing farmers in Antioquia and Magdalena, such as building disinfection stations and cement paths with farmers in Antioquia having higher adoption rates. These differences are likely due to differences in purchasing power since there were no statistically significant differences in awareness of TR4 or in the importance of disinfection.

Different actors from the public and private sector are collaborating to support banana producers in their efforts to combat TR4. It is important to carry out these prevention measures as they are the main means by which the effects of TR4 can be mitigated, given that, in Colombia, current banana plantations consist primarily of the Cavendish cultivar, which is highly susceptible to TR4. Trials of tolerant cultivars–including FHIA hybrids and Cavendish clones like GCTCV-218 and GCTCV-119 –are being carried out to protect future banana crops and ensure sustainable production [10, 56, 57]. Despite significant research efforts to identify TR4 resistant banana cultivars, commercially available dessert bananas with agronomic traits comparable to Cavendish and matching Cavendish's tolerance to Foc Race 1 remain elusive. Bred hybrids, such as the FHIA lines, often exhibit non-ideal fruit characteristics or lower yields, hindering their large-scale adoption for export [10]. Similarly, although Cavendish somaclones like 'Formosana' (GCTCV-218) demonstrate intermediate TR4 tolerance, they may not meet the stringent quality standards of international markets [10].

Notable is that this study covers only some prevention strategies. Dita et al. [58] argue that intercontinental movement of the pathogen is influenced by anthropogenic factors, such as contaminated planting material, boots, or tools. They compare this to the spread of Foc R1, where asymptomatic infected material was transported by humans, and devastated the Gros Michel cultivator of bananas in the 1900s. Despite biosecurity controls, our results and the literature review in Dita et al. [58] show that producers often neglect daily prevention strategies, increasing the risk of TR4 spreading. Although the prevention protocols established by ICA [59] and adopted in Colombia in March of 2024 [41] include ensuring good crop nutrition, maintaining drainage networks and water conduction channels, reducing irrigation in banana plantations, and avoiding the disposal of plant residues in water sources and drainage channels [59], disinfection stations represent the most frequently adopted prevention strategy in our study. The literature underscores the critical nature of managing TR4 through various stages of intervention. Martínez-Solórzano et al. [60] outline a multi-tiered approach to disease management: the application of integrated pest management for containment in the short term; the use of resistant clones in the medium term; and the implementation of clones through genetic solutions in the long term. Pérez-Matos et al. [61] advocate for the use of phytosanitary geospatial analysis to enhance preventive actions against TR4. This innovative approach could further refine our understanding of risk and guide the strategic implementation of biosecurity measures. Moreover, Rodríguez et al. [21] discuss susceptibility mapping of Colombian Musaceae lands, highlighting the importance of agro-environmental conditions, such as soil management practices, in disease spread and management. Such tools could be invaluable for policymakers and farmers to prioritize areas for intervention. Complementing Rodríguez et al. [21], Rodríguez et al. [20] provide valuable insights into the environmental and soil factors influencing TR4 prevalence. By evaluating the costs and NPV associated with investing in TR4 prevention measures, the CBA results underscore the substantial benefits derived from the implementation of such measures and align with the findings in previous studies [9, 35]. This consistency between our results and broader studies reinforces the financial viability of TR4 prevention strategies.

## Dispersion scenarios predict large economic losses from TR4

Estimated economic losses from the four dispersion scenarios in Magdalena and Antioquia show that the consequences of TR4 could be immense. Depending on the dispersion of TR4 (ranging from 25% to 100%), export losses are estimated to be between $62.5 and $250 million USD annually in Magdalena and between $114 and $456 million USD in Antioquia. Losses in land value are also estimated to be immense. These impacts would not only be felt in the banana value chain, but throughout the entire economy. These results support the words producers used to describe the impact of TR4 on the economy, jobs, food security, and environment, which highlight the potential widespread impacts of TR4. Moreover, the results align with the opinions of other stakeholders, such as extension agents, cooperative managers, and government agents, who are concerned that TR4 might negatively affect food security, regional availability of bananas, the environment, and the income of value chain actors.

## Limitations of the study

This study has some important limitations. First, this study did not analyze plantains–a crop which is also affected by TR4 and holds significant economic and food security importance in Colombia. Second, producers had limited knowledge regarding the costs linked to the implementation of disinfecting tools and machinery. Third, the CBA included only prevention strategies mentioned by producers. A fourth limitation is that although we had intended to

interview managers and pursued various methods to gain their confidence, producers and banana workers of farms with TR4 declined to participate in the survey. Consequently, our results do not fully capture the costs associated with managing the disease post-infection, such as costs related to eradicating plantations and costs cooperatives and the government accrue. This could lead to an underestimation of costs. This exclusion limits the scope of our findings only to pre-infection biosecurity efforts since our cost data were based on prevention strategies. Furthermore, while the surveyed producers implement biosecurity measures to prevent TR4, the actual effectiveness of these measures in preventing TR4 remains unknown. Any variation in effectiveness would impact the validity of the CBA findings.

Lastly, intangible costs and benefits were not included in the CBA, which may underestimate the full burden of this disease. Pattison et al. [44] highlights how banana growers in Australia fear TR4 due to previous devastations caused by Race 4 and Race 1, which led to decreased agricultural equity, social isolation, and loss of markets. Similarly, Aquino et al. [32] explains how in the Philippines, managing Fusarium wilt in Cavendish banana farms contributes to growers' sense of insecurity and isolation. There are also tangible and intangible benefits from TR4 mitigation that were not included in the CBA, such as increased certainty about banana farming, biodiversity, mental and physical health, reduced public costs for control of future outbreaks, and the cultural and traditional importance for families and communities to continue to grow bananas [43]. Accordingly, our results may not reflect these potential biases and uncertainties.

Acknowledging these limitations, we used different methods in the estimation of the CBA which shows consistent results.

## Future research opportunities

Addressing the above-mentioned limitations through future research will enhance the robustness of our results and enable recommendations to be tailored to both agricultural and non-agricultural households in banana-producing areas. Additionally, longitudinal studies could help in understanding the long-term effectiveness of prevention methods. A more detailed socioeconomic analysis of agricultural and non-agricultural households [55, 62, 63] and soil-level analysis [20, 21] in areas affected by TR4 would further enrich the literature and help policymakers direct investments to strengthen rural livelihoods. A tool like Gracia et al.'s [64] approach to help government officials and local leaders in Panama better understand the connections between climate-related risks, people's livelihoods, and project activities, could be adapted to the context of TR4 context to help develop additional prevention measures. More research should be done to identify TR4-resistant plant material. Field trials should be expanded to South America to find other solutions to combat TR4, such as banana varieties resistant or tolerant to TR4 and elicitor applications. To date, field trails have been conducted in Asia, Africa, Australia, and Europe [6, 56, 58, 65], so their applicability to South America is unknown. Lastly, artificial intelligence should be leveraged to be able to better detect and monitor TR4, spread awareness, and examine production responses to TR4 (e.g., [66] applied machine learning for Black Sigatoka). By integrating studies on socioeconomic factors, pathology, soil properties, climatic suitability mapping, and emerging technologies, stakeholders can identify high-risk areas and implement targeted interventions.

Weighing intangible costs and benefits presents significant challenges due to their non-quantifiable nature. However, incorporating these components is crucial for a comprehensive understanding of any analysis. Therefore, it is recommended that future studies develop and utilize methodologies specifically designed to identify, analyze, and include intangible factors.

By doing so, researchers can gain a more holistic view that captures the full spectrum of impacts, leading to more informed decision-making.

## Conclusion

This study underscores the urgent need for coordinated efforts among stakeholders, including government agencies, researchers from a variety of disciplines, and local communities, to combat the spread of TR4, safeguard the long-term sustainability of the banana industry in Colombia and beyond, and build resilience among banana producers and their communities. This study, along with the existing literature [19, 66, 67] emphasizes the need for proactive measures, including prevention, rapid containment, and quarantine protocols, to prevent the spread of TR4, as well as promoting practices to improve soil health, early detection, containment, and awareness through education about phytosanitary conditions [3, 18]. Based on the results and discussion, we provide several actionable recommendations below to help combat the spread of TR4 in Colombia. These recommendations could be evaluated and adapted to local contexts in other countries, such as neighboring Venezuela and Peru, where banana production is also threatened by TR4 [19–21].

First, the implementation of cement roads and disinfection stations is recommended. Since TR4 is a soil-borne pathogen, disinfection stations strategically located at the entrance of banana farms represent a critical line of defense: They not only prevent the introduction and transmission of the pathogen, but also help contain its spread through the rigorous disinfection of footwear in direct contact with the soil. These stations should not only be built and used to disinfect footwear, but also machinery and vehicles moving between farms. Despite some producers wanting to implement prevention measures, it is not feasible for them due to income constraints. Support for building disinfection stations and cement paths should be provided by policymakers and investors. Given that marketing companies currently support banana producers against other diseases, one could assume that they may also be interested in also supporting banana producers in combating TR4. In Colombia, public and private entities (including ICA, Agrosavia, the Ministry of Agriculture, ASBAMA, and Augura) joined together to prevent the spread of TR4. They agreed to invest nearly 7.2 million USD in TR4 prevention, surveillance, and awareness measures from 2019 to 2023 [68–76], which is estimated to benefit 1,340 mostly small-scale producers [77]. There is no data on how much will be invested in the future. Support for producers is also necessary to help them obtain required inputs to provide some financial relief and possibly free up funds for TR4 prevention.

Second, raise awareness about TR4. Non-banana farmers and laborers are unaware of TR4, so they don't think twice about trespassing on banana plantations to take bananas without undertaking shoe disinfection. Trespassing could be reduced if community members were made aware of the dire consequences of TR4. Based on the high level of awareness about TR4 among producers because of sensitization campaigns, we recommend that the government should also raise awareness about the benefits of investing in TR4 prevention methods as well as the risks of using corms as seeds. Awareness campaigns about the importance of using certified banana seeds would help guarantee the quality of the plant material and reduce the risk of TR4 infection. Prevention protocols must be more widely communicated among producers, so that they implement additional preventive measures–beyond the few identified in this study–to reduce the risk of TR4. Producers need practical actions at the ready in case their farms are infected with TR4 [2].

Third, the use of technology, such as digital communication channels, mobile apps, and drones, should be expanded to spread awareness and help detect and prevent TR4. Digital communication channels, such as WhatsApp, serve as crucial platforms for knowledge sharing

and collaboration. These channels may prove useful to reduce the negative consequences of TR4 since producers may take advantage of these digital tools to share their experiences dealing with TR4. Technology has the potential to improve crop management, disease prevention and management, and information access and sharing. Drones could be used to identify symptomatic or suspect plants as well as areas prone to TR4 that deserve further investigation through other techniques [78]. Nevertheless, drone usage is low in Colombia because it is expensive and relatively new, so it is not considered as an option by most. Cooperatives and associations could pool their resources together to increase the use of drones. The government should also conduct monitoring activities, such as via drones, to detect symptoms of TR4 on plantations. Mobile apps, such as Tumaini [79], that utilize artificial intelligence also hold great potential for detecting TR4 in banana plants.

Lastly, the government should invest in diversifying and strengthening crops resistant to TR4 and resources in rural areas to enhance resilience and support sustainable land management practices [62, 80]. Producers should be encouraged to grow crops, such as paddy rice, sugarcane, tapioca, onion, and pineapple, which are non-host crops for TR4 and reduce the inoculum level of the pathogen [2].

The recent emergence of TR4 in Colombia, Peru, and Venezuela underscores the urgency for effective containment and control measures to safeguard the banana industry and mitigate potential economic losses [19–21]. Producer associations and cooperatives are key in supporting banana producers as they are the closest institution to farmers and support them in different aspects of banana production, such as obtaining less expensive inputs, providing technical assistance, and–recently with TR4 prevention and surveillance–by facilitating the application of prevention measures on banana farms. Producer associations and cooperatives should be involved in each of the above recommendations. A multipronged approach is needed to prevent TR4 from spreading further in Colombia. Increasing awareness, expanding currently used approaches, adopting new technology, and finding banana varieties resistant or tolerant to TR4 is imperative. This study has shown that the benefits of investing in prevention measures largely outweigh the costs.

## Supporting information

**S1 Table. Mean t-tests for comparing averages between departments.**
(DOCX)

## Acknowledgments

Special thanks to Jorge Eliecer Vargas who provided valuable guidance and information during the socioeconomic study. We are thankful for the support in the data collection process to the banana producer associations Augura and ASBAMA, the producer cooperatives, members from the Latin American and Caribbean Network of Fair-Trade Small Producers and Workers (CLAC, for its acronym in Spanish) and to Leonardo García and his data collection team. The opinions expressed are those of the authors and do not necessarily reflect those of the funders or institutions of affiliation.

## Author Contributions

**Conceptualization:** Thea Ritter, Edward Martey, Jonathan Mockshell.

**Data curation:** Diego Álvarez, Leslie Estefany Mosquera.

**Formal analysis:** Diego Álvarez, Leslie Estefany Mosquera.

**Funding acquisition:** Jonathan Mockshell.

**Investigation:** Diego Álvarez, Leslie Estefany Mosquera.

**Methodology:** Thea Ritter, Leslie Estefany Mosquera, Edward Martey, Jonathan Mockshell.

**Project administration:** Jonathan Mockshell.

**Resources:** Jonathan Mockshell.

**Supervision:** Thea Ritter, Jonathan Mockshell.

**Validation:** Edward Martey.

**Writing – original draft:** Thea Ritter, Diego Álvarez, Leslie Estefany Mosquera, Jonathan Mockshell.

**Writing – review & editing:** Thea Ritter, Edward Martey.

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
