## [Decision Letter · Decision Letter 0]

14 May 2024

PONE-D-24-08984A socioeconomic and cost benefit analysis of Tropical Race 4 (TR4) among producers in ColombiaPLOS ONE

Dear Dr. Ritter,

Thank you for submitting your manuscript to PLOS ONE. After careful consideration, we feel that it has merit but does not fully meet PLOS ONE’s publication criteria as it currently stands. Therefore, we invite you to submit a revised version of the manuscript that addresses the points raised during the review process.

We look forward to receiving your revised manuscript.

Kind regards,

Rajappa Janyanaik Joga, PhD

Academic Editor

PLOS ONE

Journal Requirements:

"This work was funded by the Deutsche Gesellschaft für Internationale Zusammenarbeit (GIZ) in the project, “Innovations for the prevention and management of the banana fungal disease Foc TR4 (ALER4TA)”, implemented by the Fund for the Promotion in Agriculture (i4Ag) as part of the special initiative Transformation of Agricultural and Food Systems."

"Special thanks to Jorge Eliecer Vargas who provided valuable guidance and information during the socioeconomic study. We are grateful for the funding for this work from the Deutsche Gesellschaft für Internationale Zusammenarbeit (GIZ). We are thankful for the support in the data collection process to the banana producer associations Augura and Asbama, the producer cooperatives, members from the Latin American and Caribbean Network of Fair-Trade Small Producers and Workers (CLAC, for its acronym in Spanish) and, last but not least, to Leonardo  García and his data collection team. The opinions expressed are those of the authors, and do not necessarily reflect those of the funders or institutions of affiliation."

"This work was funded by the Deutsche Gesellschaft für Internationale Zusammenarbeit (GIZ) in the project, “Innovations for the prevention and management of the banana fungal disease Foc TR4 (ALER4TA)”, implemented by the Fund for the Promotion in Agriculture (i4Ag) as part of the special initiative Transformation of Agricultural and Food Systems."

6. In the online submission form, you indicated that "Not all data can be shared publicly because of concerns of confidentiality. Some data are available from the authors (contact via Jonathan Mockshell at j.mockshell@cgiar.org) for researchers who meet the criteria for access to the data."

7. Please amend either the title on the online submission form (via Edit Submission) or the title in the manuscript so that they are identical.

8. We note you have included a table to which you do not refer in the text of your manuscript. Please ensure that you refer to Table 5 in your text; if accepted, production will need this reference to link the reader to the Table.

9. We notice that your supplementary tables are included in the manuscript file. Please remove them and upload them with the file type 'Supporting Information'. Please ensure that each Supporting Information file has a legend listed in the manuscript after the references list.

Reviewers' comments:

Reviewer's Responses to Questions

**Comments to the Author**

1. Is the manuscript technically sound, and do the data support the conclusions?

Reviewer #1: Partly

Reviewer #2: No

Reviewer #3: No

Reviewer #4: Partly

2. Has the statistical analysis been performed appropriately and rigorously? 

Reviewer #1: Yes

Reviewer #2: No

Reviewer #3: No

Reviewer #4: I Don't Know

3. Have the authors made all data underlying the findings in their manuscript fully available?

Reviewer #1: Yes

Reviewer #2: No

Reviewer #3: Yes

Reviewer #4: Yes

4. Is the manuscript presented in an intelligible fashion and written in standard English?

Reviewer #1: Yes

Reviewer #2: Yes

Reviewer #3: Yes

Reviewer #4: Yes

5. Review Comments to the Author

Reviewer #1: Scientific Quality:

The paper presents an important contribution to the literature by investigating the socioeconomic implications of Fusarium oxysporum f. sp. cubense Tropical Race 4 (TR4) on banana producers in Colombia. The study employs a mixed-methods approach, including interviews with producers and key stakeholders, as well as a cost-benefit analysis, to assess the susceptibility of banana farming systems to TR4. The findings shed light on the financial viability of current monitoring strategies and provide valuable insights into the effectiveness of preventive measures implemented by producers.

While the study provides a comprehensive analysis of the socioeconomic aspects of TR4 on banana producers in Colombia, the methodology section lacks detail regarding the sampling strategy for selecting interview participants. Providing information on how participants were chosen would enhance the transparency and reproducibility of the study.

The paper would benefit from a more thorough discussion of the limitations inherent in the cost-benefit analysis. For instance, the analysis may overlook certain intangible costs and benefits associated with TR4 mitigation measures, such as the psychological impact on farmers or the long-term environmental consequences. For example:

Line 652: The study of Martínez et al. (2023) and Olivares (2023) underscores the critical threat posed by TR4 to the global banana industry, particularly in Latin America and the Caribbean (LAC), where its presence has been increasingly reported. Indeed, as highlighted by Rodríguez et al. (2023a; 2023b), Campos (2023) TR4 represents a significant risk to banana production in LAC countries, including Colombia, Peru, and Venezuela. The recent emergence of TR4 in these regions underscores the urgency for effective containment and control measures to safeguard the banana industry and mitigate potential economic losses.

Moreover, the socioeconomic characterization of banana-producing regions, such as the Changuinola district in Panama (Rodríguez et al., 2021; Gracia et al. 2020; Montenegro et al. 2021), serves as a valuable reference for understanding the socioeconomic dynamics and vulnerability factors that may exacerbate the impact of TR4 on local communities. Identifying key socio-economic indicators, such as education, unemployment, and food security, enables stakeholders to formulate tailored strategies to enhance resilience and support sustainable land management practices (Olivares et al. 2020; Olivares and Hernández, 2020).

Furthermore, Vega et al. (2022); Olivares et al. (2022); and Campos (2023), emphasize the need for proactive measures, including prevention, rapid containment, and quarantine protocols, to mitigate the spread of TR4 and safeguard the livelihoods of small and medium-sized banana farmers in affected regions. By integrating risk analysis and climatic suitability mapping, stakeholders can identify high-risk areas and implement targeted interventions to prevent further dissemination of the pathogen. The findings presented in the paper underscore the urgent need for coordinated efforts among stakeholders, including government agencies, researchers, and local communities, to combat the spread of TR4 and safeguard the long-term sustainability of the banana industry in Colombia and beyond.

The abstract provides a concise overview of the study; however, it could be improved by including specific quantitative results from the cost-benefit analysis to give readers a better understanding of the magnitude of potential losses and benefits associated with TR4 mitigation strategies.

Recommendations:

Consider providing more detailed information on the sampling methodology employed for selecting interview participants, including criteria for inclusion and exclusion, to enhance the rigor and reproducibility of the study.

Expand the discussion section to include a more thorough analysis of the limitations of the cost-benefit analysis, including potential biases and uncertainties, to provide a balanced interpretation of the findings.

Improvement Advice:

To strengthen the scientific rigor of the study, consider incorporating quantitative measures of the socioeconomic impact of TR4 on banana producers, such as estimating the potential reduction in yields or income losses due to TR4 infection.

To enhance the generalizability of the findings, consider expanding the study to include a broader sample of banana-producing regions in Colombia and possibly other countries affected by TR4, to capture variations in socioeconomic conditions and farming practices.

References:

Incorporating the suggested and relevant recent literature is crucial for enriching the scholarly discourse surrounding the socioeconomic impact of Fusarium wilt (TR4) on banana production in Colombia and neighboring regions. By drawing upon the latest research findings and insights from impact journals, such as those focusing on banana pathology, agricultural economics, and regional socioeconomic dynamics, the manuscript can offer a comprehensive and up-to-date analysis of the challenges and opportunities faced by banana producers. Furthermore, integrating recent literature enhances the paper's credibility, ensuring that it aligns with the latest advancements and understanding in the field. This comprehensive approach not only strengthens the scientific foundation of the study but also provides valuable context and perspective for policymakers, researchers, and stakeholders seeking effective strategies to mitigate the impact of TR4 on banana production and safeguard the livelihoods of farming communities.

Campos, B. O. O. (2023). Banana production in Venezuela: Novel solutions to productivity and plant health. Springer Nature. https://doi.org/10.1007/978-3-031-34475-6

Gracia, E. J. M., Rodríguez, J. E. P., & Campos, B. O. O. (2020). Adaptation to climate change in indigenous food systems of the Teribe in Panama: a training based on Cristal 2.0. Revista Luna Azul, (51), 182-197. : https://doi.org/10.17151/luaz.2020.51.10

Martínez, G.; Olivares, B.O.; Rey, J.C.; Rojas, J.; Cardenas, J.; Muentes, C.; Dawson, C. 2023. The Advance of Fusarium Wilt Tropical Race 4 in Musaceae of Latin America and the Caribbean: Current Situation. Pathogens, 12, 277. https://doi.org/10.3390/pathogens12020277

Montenegro-Gracia, E. J., Pitti-Rodríguez, J. E., & Olivares-Campos, B. O. (2021). Identification of the main subsistence crops of Teribe: a case study based on multivariate techniques. Idesia (Arica), 39(3), 83-94. http://dx.doi.org/10.4067/S0718-34292021000300083

Olivares B, Vega A, Calderón MAR, Rey JC, Lobo D, Gómez JA, Landa BB. 2022. Identification of Soil Properties Associated with the Incidence of Banana Wilt Using Supervised Methods. Plants, 11(15):2070. https://doi.org/10.3390/plants11152070

Olivares, B., Hernández, R. 2020. Application of multivariate techniques in the agricultural land’s aptitude in Carabobo, Venezuela. Tropical and Subtropical Agroecosystems, 23(2):1-12. http://dx.doi.org/10.56369/tsaes.3233

Olivares, B., Pitti, J., Montenegro, E. 2020. Socioeconomic characterization of Bocas del Toro in Panama: an application of multivariate techniques. Revista Brasileira de Gestao e Desenvolvimento Regional, 16(3):59-71. https://doi.org/10.54399/rbgdr.v16i3.5871

Olivares, B.O. (2023). Fusarium Wilt of Bananas: A Threat to the Banana Production Systems in Venezuela. In: Banana Production in Venezuela. The Latin American Studies Book Series. Springer, Cham. https://doi.org/10.1007/978-3-031-34475-6_3

Rodríguez, J. E. P., Olivares, B. O., Montenegro, E. J., Miller, L., & Ñango, Y. (2021). The role of agriculture in the Changuinola District: A case of applied economics in Panama. Tropical and Subtropical Agroecosystems, 25(1). http://dx.doi.org/10.56369/tsaes.3815

Rodríguez-Yzquierdo, G.; Olivares, B.O.; González-Ulloa, A.; León-Pacheco, R.; Gómez-Correa, J.C.; Yacomelo-Hernández, M.; Carrascal-Pérez, F.; Florez-Cordero, E.; Soto-Suárez, M.; Dita, M.; et al. Soil Predisposing Factors to Fusarium oxysporum f.sp Cubense Tropical Race 4 on Banana Crops of La Guajira, Colombia. Agronomy 2023, 13, 2588. https://doi.org/10.3390/agronomy13102588

Rodríguez-Yzquierdo, G.; Olivares, B.O.; Silva-Escobar, O.; González-Ulloa, A.; Soto-Suarez, M.; Betancourt-Vásquez, M. Mapping of the Susceptibility of Colombian Musaceae Lands to a Deadly Disease: Fusarium oxysporum f. sp. cubense Tropical Race 4. Horticulturae 2023, 9, 757. https://doi.org/10.3390/horticulturae9070757

Vega, A.; Olivares, B.O.; Rueda Calderón, M.A.; Montenegro-Gracia, E.; Araya-Almán, M.; Marys, E. 2022. Prediction of Banana Production Using Epidemiological Parameters of Black Sigatoka: An Application with Random Forest. Sustainability 14, 14123. https://doi.org/10.3390/su142114123

Reviewer #2: The article is not solid, the data obtained is hypothetical because the producers interviewed do not have the disease and therefore, there is no direct relationship between the data and the conclusions. For the cost benefit analysis, they used very general cost data that does not come from a reliable source.

The sample size is not adequate, it is unbalanced in the study areas. The data obtained does not allow us to have an analysis of costs, much less the benefits of the implementation of biosecurity vs. the losses that can be caused by the disease that is the focus of the study, TR4. If anything, they can serve to characterize the producers of Magalenda, because they do not even have a sample population that allows them to characterize the producers of Antioquia.

The title of the article has nothing to do with the results and furthermore the conclusions reached have nothing to do with the information obtained in the materials and methods, reflecting a very large decontextualization of the TR4 problem in Colombia.

Although the authors explain in the text why they do not have information on the farms affected by TR4, the basic information exercise was also very poor, for example, the production costs of bananas for export, if they could be obtained with good work in field and then from this information it would have been possible to estimate the cost of biosecurity implementation.

The cost of the disease has many dimensions, for example in: loss of jobs, or loss of land value, that were not explored by the authors. On the other hand, it is strange in the document that it was not analyzed what it has cost the country and its institutions to prevent and carry out surveillance on the disease, these data could also be estimated and no effort was made in this regard.

It is not an article that contributes to knowledge and is not interesting. It does not have rigor in the materials and methods, nor in the recommendations and conclusions, which are not derived from the results obtained, rather they are hypotheses, which also demonstrate a low contextualization and knowledge of the topic and the reality of the disease in Colombia and other South American countries.

The recommendations for the prevention of the pathogen indicate a great lack of knowledge of phytopathology in general and of the understanding of the sociocultural patterns of Colombia and Latin America and the Caribbean.

Reviewer #3: The article is not solid, mainly because information was obtained from farms that do not have Foc TR4, the data obtained from free farms makes the results hypothetical. The estimated costs are not adequate because they do not consider real costs of farms that implement biosecurity measures in the presence of the pathogen. The population sampled between both departments is very unbalanced and the analysis for the department of Antioquia and its distribution of types of producers is not representative.

The title of the article has no relationship with the results and conclusions. There is a marked decontextualization of the Foc TR4 problem in Colombia and the productive realities of export bananas in that country. Socioeconomic impact indicators such as job losses are not considered in this study and that is significant for these regions that make a living from banana production.

The article is not robust and does not represent new knowledge, due to the hypothetical nature of the results and its poor capacity for analysis.

Reviewer #4: General comments:

The paper by Ritter et al is an important assessment of the impact of TR4 in commercial and small-scale farming systems in Colombia. The authors could only work with the situation available, which includes the limited spread of the disease in the study areas, and the collaboration of growers. Still, the information obtained is important for the larger scientific community, especially the outcome that investment in preventative and mitigating measures can significantly increase the net value of banana production in both regions. The paper is reasonably well written, but the Discussion can be more comprehensive with more references.

I also have some specific questions and recommendations, which are listed below.

Title:

Why does the title of the manuscript on page 1 differ from that on page 3?

Abstract:

Page 3, line 46: ‘Susceptibility’ should rather be replaced with ‘vulnerability’. Also add the region to this sentence, i.e. Colombia.

Introduction:

Page 4, line 66: Replace ‘targets’ with ‘blocks’.

Page 4, line 70: Please indicate what you mean with ‘but also raises concerns for various other crops’.

Page 4, line 70: What is meant with ‘Foc exhibits diverse populations’?

Page 4, line 70: In the same sentence it is stated that ‘TR4 being the sole strain capable of affecting the economically vital Cavendish cultivar’, which is not entirely true. Fusarium subtropical race 4 can also affect Cavendish bananas, although this happens in the subtropics. Also, do the authors consider Foc VCG 0121 a member of Foc TR4, as this strain can also cause disease to Cavendish bananas.

Page 4, line 75: There is no proof that Foc TR4 is more aggressive than other races. It only affects more banana cultivars than the other races.

Pg 4, line 79: TR4 was not identified in Asia in the 1970s. That is simply speculation. The fungus was only identified after affecting Cavendish bananas in Malaysia and Indonesia in the 1990’s.

Pg 4, line 80: Please provide references for the occurrence of Foc TR4 in Africa, Colombia, Peru and Venezuela.

Pg 4, line 88: Please provide a reference for the interventions implemented in Colombia.

Pg 5, lines 92-93: Please see de Figueiredo Silva et al. (2023). Estimating worldwide benefits from improved bananas resistant to Fusarium Wilt Tropical race 4. Journal of the Agricultural and Applied Economics Association 2(1).

Materials and Methods:

Page 6, lines 130-131: The sentence starting with ‘This section ….’ is unnecessary. Please remove.

Page 6, lines 133-134. Rewrote this as follows: The departments of Antioquia and Magdalene in Colombia were selected due to several strategic considerations (Fig. 1).

Page 6, line 134. Please start a sentence with a capital letter. It would also be acceptible if you start the sentence with ‘Firstly, these departments ….’.

Page 7, line 157: Mention the name of your organization here.

Page 8, line 173: ‘….as explained below.’

Page 10, line 244: Please correct this sentence by deleting one ‘steps’.

Page 12, line 260: This sentence starting with ‘This section …’ is unnecessary, and can be deleted.

Page 15, line 329: Should the mean yield in Antioquia not be 46.24 instead of 44.1 tons/ha?

Page 16, line 341: The word ‘conditions’ can possibly be replaced with ‘differences’.

Page 16, line 353: Fix $688 USD.

Page 17, line 369: In section 3.2., more reference should be made to Table A1.

Page 19, line 414: In section 3.2.2., please briefly describe the nature of the cement paths and fences. Are the paths between or in plantations, between blocks? Are the fences on farm borders, or within farms?

Page 21, lines 459-464: Not necessary to repeat this information that was already mentioned in the Methodology section.

Page 21, line 471: Table 3 provides a comprehensive overview ….

Pages 21, 22 and 23, sections 3.3.1., 3.3.2. and 3.3.3: Too much Methodology included in the beginning of these sections. Rather move the methods used to the Methodology sections.

Page 23, line 511: This must be a mistake, as I cannot see the financial performance of biosecurity strategies being presented in Table 1. Should it be Table 5?

Page 24, line 531: Refer to Table 5 in this section.

Page 25, line 551. Refer to Table 6 here.

Discussion:

Page 27, lines 575-578: The first two sentences in this paragraph says essentially the same thing. Please merge or delete one of the sentences.

Page 27, line 578: The statement starting with ‘TR4 poses a serious threat ….’ is redundant and should rather be used in the Introduction.

Page 27, line 585: Please add references to this paragraph.

Page 27, lines 586-588: The authors should vindicate the statement ‘banana farms …. in both Antioquia and Magdalena are highly susceptible to the spread of TR4’ based on the findings of your study.

Page 27, line 588: Infested soil, not infected soil.

Page 28, lines 594-605: This paragraph is full of unsupported statements, such as that TR4 spread would lead to massive job losses, unemployment, and hunger. Why can these farms not grow other crops, for instance, to generate income? Are there any literature or experiences substantiating these statements? Maybe cite some work done in the early 1900s in Latin America following the Gros Michel – race 1 epidemic.

Page 28, line 606: This section also requires some references for certain statements, especially on pro-active measures as the main means by which TR4 can be mitigated, and on the availability of resistant varieties. Please add these.

Page 29, line 614: It is not true that there are no banana varieties resistant or tolerant to TR4. Both FHIA-1 and GCTCV-218 (Formosana) are resistant and tolerant, respectively. Consult the literature on this. There might be other reasons why these are not acceptable for planting in Colombia.

Page 29, lines 615-617: What do these statistical regional differences mean? This should also be discussed here, and not only the reason for the differences.

Page 30, line 642: I appreciate that the current study is very unique, and based on predictions. If possible, however, one hope that the authors could compare their results to what happened in terms of the financial and social impact during the Gros Michel-Foc race 1 epidemic in Latin America in the early 1900s.

Page 31, line 666 onwards: There are also other interventions that need to be mentioned here. For instance, the fencing of properties and regulation of access, the digging of trenches and canals to direct the flow of water. Disinfection is a science by its own, and should be correctly used, while cement paths are not affordable by all.

Page 31, line 679-683: The authors also discuss what government, cooperatives and associations should do, but not what they have already done, or not done. That would provide some perspective.

Pg 32, line 704: The use of drones to detect TR4 is not well proved and should be recommended with care. Rather explain that it should be used to identify symptomatic or suspect plants that must be further investigated.

6. PLOS authors have the option to publish the peer review history of their article (what does this mean?). If published, this will include your full peer review and any attached files.

Reviewer #1: No

Reviewer #2: No

Reviewer #3: No

Reviewer #4: No

---

## [Author Response · Author response to Decision Letter 0]

26 Jul 2024

Review Comments to the Author

Reviewer #1: 

Scientific Quality:

-The paper presents an important contribution to the literature by investigating the socioeconomic implications of Fusarium oxysporum f. sp. cubense Tropical Race 4 (TR4) on banana producers in Colombia. The study employs a mixed-methods approach, including interviews with producers and key stakeholders, as well as a cost-benefit analysis, to assess the susceptibility of banana farming systems to TR4. The findings shed light on the financial viability of current monitoring strategies and provide valuable insights into the effectiveness of preventive measures implemented by producers.

Thank you for taking the time to read our manuscript and provide valuable feedback and comments. We appreciate your time and dedication to improving our manuscript and thus knowledge base in the scientific community. We have replied to each of your comments below, with line numbers referring to those in the version with tracked changes. 

-While the study provides a comprehensive analysis of the socioeconomic aspects of TR4 on banana producers in Colombia, the methodology section lacks detail regarding the sampling strategy for selecting interview participants. Providing information on how participants were chosen would enhance the transparency and reproducibility of the study.

Thank you for bringing this to our attention. We added more detail to the paper in terms of the sampling strategy (lines 214-223). To summarize, all banana producers who visited cooperatives when the enumeration team was present and all banana producers encountered in villages where the cooperative leaders took the enumeration team were asked to participate in the survey. However, about 60% of producers approached in cooperatives and 50% approached in the field declined to participate in our study. Despite explaining the confidentiality of participation, producers still feared that their economic standing and ability to take out loans would be affected through their participation in our study.

- The paper would benefit from a more thorough discussion of the limitations inherent in the cost-benefit analysis. For instance, the analysis may overlook certain intangible costs and benefits associated with TR4 mitigation measures, such as the psychological impact on farmers or the long-term environmental consequences. For example:

Line 652: The study of Martínez et al. (2023) and Olivares (2023) underscores the critical threat posed by TR4 to the global banana industry, particularly in Latin America and the Caribbean (LAC), where its presence has been increasingly reported. Indeed, as highlighted by Rodríguez et al. (2023a; 2023b), Campos (2023) TR4 represents a significant risk to banana production in LAC countries, including Colombia, Peru, and Venezuela. The recent emergence of TR4 in these regions underscores the urgency for effective containment and control measures to safeguard the banana industry and mitigate potential economic losses.

Moreover, the socioeconomic characterization of banana-producing regions, such as the Changuinola district in Panama (Rodríguez et al., 2021; Gracia et al. 2020; Montenegro et al. 2021), serves as a valuable reference for understanding the socioeconomic dynamics and vulnerability factors that may exacerbate the impact of TR4 on local communities. Identifying key socio-economic indicators, such as education, unemployment, and food security, enables stakeholders to formulate tailored strategies to enhance resilience and support sustainable land management practices (Olivares et al. 2020; Olivares and Hernández, 2020).

Furthermore, Vega et al. (2022); Olivares et al. (2022); and Campos (2023), emphasize the need for proactive measures, including prevention, rapid containment, and quarantine protocols, to mitigate the spread of TR4 and safeguard the livelihoods of small and medium-sized banana farmers in affected regions. By integrating risk analysis and climatic suitability mapping, stakeholders can identify high-risk areas and implement targeted interventions to prevent further dissemination of the pathogen. The findings presented in the paper underscore the urgent need for coordinated efforts among stakeholders, including government agencies, researchers, and local communities, to combat the spread of TR4 and safeguard the long-term sustainability of the banana industry in Colombia and beyond.

Thank you very much for this very detailed recommendation. Below, we address the points made above. 

We have incorporated the aspect of intangible costs and benefits throughout the manuscript, beginning in the methods section where we explain that while CBA can help stakeholders make rational decisions in complex situations, quantifying intangible benefits, such as through an index or monetizing intangible benefits, is difficult and subjective (Kumar et al., 2021; Wun et al., 2022) (lines 230-236). In an attempt to capture these intangible aspects that the cost-benefit analysis is unable to (Hamundu et al., 2012), producers were asked about the impact of TR4 on different aspects of their lives and community, namely on the economy, jobs, food security, and the environment. Words which respondents used highlight some of the intangible impacts of TR4. We did not include this in the previous version of the manuscript, but have now added this qualitative information to the revised version. These qualitative results supplement the quantitative CBA (lines 280-287) and the results are now presented in a new subsection “Intangible aspects of TR4” (lines 766-778), which is also now discussed in the Discussion section (lines 944-954). 

Regarding the cost-benefit analysis, we admit that our analysis is based on partial budgeting which does not fully capture all the costs including the intangible costs and benefits associated with TR4 prevention measures and thus may underestimate the full burden of TR4 (lines 927-928). For example, there are intangible costs of TR4 such as fear, agricultural equity, social inclusion, loss of markets, insecurity, and isolation (Aquino et al., 2012; Pattison et al., 2013) (lines 928-932). In addition, there are intangible benefits of TR4 mitigation, such as increased certainty about banana farming, biodiversity, mental and physical health, reduced public costs for control of future outbreaks, and the cultural and traditional importance for families and communities to continue to banana farm (Wun et al., 2022) (lines 932-937). We now discuss these at the end of the subsection “Limitations of the study” towards the end of the Discussion section. 

Despite these limitations, we used different methods in the estimation of the cost-benefit analysis which shows consistent results. We also acknowledge that valuing intangible costs and benefits can be extremely difficult. It is recommended that future studies include methodologies that allow these intangible aspects to be analyzed, identified and included. This suggestion was added in “Future research opportunities” section of the manuscript (see lines 962-967). 

We have now included the studies you mentioned that highlight the risk of TR4 to LAC countries in the second paragraph of the manuscript (namely, Campos, 2023; Martínez et al., 2023; Rodríguez et al., 2023a; 2023b, Olivares Campos, 2023) (lines 95-96 and 101-104). 

Your example and the literature you discussed on the risk of TR4 in Latin America was added to the discussion in the subsection “Vulnerability of banana farming systems and households to TR4” (lines 773-812).

We added a discussion of other strategies identified and recommended in the literature for the management of TR4 in the subsection “Current monitoring strategies and their financial viability” (lines 813-864). 

We incorporated the importance of understanding key socioeconomic dynamics and vulnerability factors (Montenegro et al. 2021; Olivares et al., 2020; Rodríguez et al., 2021) (lines 933-948) and the urgency for effective containment measures in the Conclusion section (lines 973-977).

We liked the second to last and last sentence of your above comment so much, that we rephrased them and included them as the first two sentences of our Conclusion. 

Thank you again for your very valuable comments and advice! 

-The abstract provides a concise overview of the study; however, it could be improved by including specific quantitative results from the cost-benefit analysis to give readers a better understanding of the magnitude of potential losses and benefits associated with TR4 mitigation strategies.

Thank you. We added specific quantitative results to the Abstract, namely the net present value and benefit-cost ratio from the cost-benefit analysis. 

Recommendations:

-Consider providing more detailed information on the sampling methodology employed for selecting interview participants, including criteria for inclusion and exclusion, to enhance the rigor and reproducibility of the study.

Thank you for this suggestion. In the initial phase, we met with government officials and experts in Colombia’s banana sector to determine which actors to interview (lines 208-210). To obtain a wide range of perspectives, we decided to interview a variety of stakeholders. Our proposed sample included actors both with and without TR4 on their banana farms (lines 230-231). However, despite several months of trying to interview producers with TR4, we were unable to interview producers (or managers or banana workers) of farms with TR4 due to their unwillingness to participate in the survey (this was elaborated on in the previous version of the manuscript, starting on line 231 in the revised version). Given our response and changes based on this comment and your second comment on the sampling strategy (which we addressed on lines 214-223), we believe that we have provided enough detail now on the sampling strategy. 

-Expand the discussion section to include a more thorough analysis of the limitations of the cost-benefit analysis, including potential biases and uncertainties, to provide a balanced interpretation of the findings.

Thank you for this suggestion. In addition to the above-mentioned inclusion of the limitation of the CBA of not including intangible benefits and costs (lines 932-937), we also now include a discussion of other limitations of the CBA in the newly added subsection “Limitations of the study” (starting on line 904). For example, we elaborate that our results should be interpreted with caution given potential biases and uncertainties. In addition, we specify that the full economic impact of TR4 – especially post-infection – was not taken into account. We also acknowledge in the manuscript that future studies should include data from affected farms and would also benefit from longitudinal data. Despite trying for over four months to gather data from producers with TR4 present on their banana farms, we were unable to interview them because they were not willing to share information with us. While we recommend that future studies gather data from these producers, we don’t have any clear advice on how researchers can practically do this given the extreme difficulties we faced interviewing this subset of producers. 

Improvement Advice:

-To strengthen the scientific rigor of the study, consider incorporating quantitative measures of the socioeconomic impact of TR4 on banana producers, such as estimating the potential reduction in yields or income losses due to TR4 infection.

Thank you for this very valuable suggestion. We added a table (Table 5, line 735) that includes estimates of the annual impact of TR4 at the department-level. We show impacts on the number of hectares, banana production, direct and indirect employment, export losses, and the loss of land value. In addition, we show impacts at the producer-level in Table 6 (line 745) on affected areas, banana production, export income, domestic sales income, and land value for the average producer in each department. 

-To enhance the generalizability of the findings, consider expanding the study to include a broader sample of banana-producing regions in Colombia and possibly other countries affected by TR4, to capture variations in socioeconomic conditions and farming practices.

Thank you for this comment. The two study regions were chosen for very specific reasons: they are the top two banana producing departments in Colombia; they have distinct farming systems so we could compare farming systems; they have different banana markets (export vs. domestic bananas); they mostly plant with Cavendish cultivators to produce export bananas; and one department (Magdalena) had three confirmed cases of TR4 when the study took place (lines 166-179). At the time of data collection, we were not able to conduct the study in other regions of Colombia or in other countries because of funding constraints. We had initially set out to conduct interviews in another department, La Guajira, which had confirmed cases of TR4 on large banana plantations; however, managers of these large plantations were unwilling to speak with enumerators, the authors, and hired consultants, let alone be interviewed, so we had to withdraw La Guajira from our study area. In addition to the already related and cited studies that took place in other countries, we have added the references you cited below to provide more generalizability of our findings. For example, in the Conclusion (lines 979-982) we state that the recommendations could be evaluated and adapted to local contexts in other countries, such as neighboring Venezuela and Peru, where banana production is threatened by TR4 (Campos, 2023; Rodríguez et al., 2023a, 2023b).

References:

-Incorporating the suggested and relevant recent literature is crucial for enriching the scholarly discourse surrounding the socioeconomic impact of Fusarium wilt (TR4) on banana production in Colombia and neighboring regions. By drawing upon the latest research findings and insights from impact journals, such as those focusing on banana pathology, agricultural economics, and regional socioeconomic dynamics, the manuscript can offer a comprehensive and up-to-date analysis of the challenges and opportunities faced by banana producers. Furthermore, integrating recent literature enhances the paper's credibility, ensuring that it aligns with the latest advancements and understanding in the field. This comprehensive approach not only strengthens the scientific foundation of the study but also provides valuable context and perspective for policymakers, researchers, and stakeholders seeking effective strategies to mitigate the impact of TR4 on banana production and safeguard the livelihoods of farming communities.

Campos, B. O. O. (2023). Banana production in Venezuela: Novel solutions to productivity and plant health. Springer Nature. https://doi.org/10.1007/978-3-031-34475-6

Gracia, E. J. M., Rodríguez, J. E. P., & Campos, B. O. O. (2020). Adaptation to climate change in indigenous food systems of the Teribe in Panama: a training based on Cristal 2.0. Revista Luna Azul, (51), 182-197. : https://doi.org/10.17151/luaz.2020.51.10

Martínez, G.; Olivares, B.O.; Rey, J.C.; Rojas, J.; Cardenas, J.; Muentes, C.; Dawson, C. 2023. The Advance of Fusarium Wilt Tropical Race 4 in Musaceae of Latin America and the Caribbean: Current Situation. Pathogens, 12, 277. https://doi.org/10.3390/pathogens12020277

Montenegro-Gracia, E. J., Pitti-Rodríguez, J. E., & Olivares-Campos, B. O. (2021). Identification of the main subsistence crops of Teribe: a case study based on multivariate techniques. Idesia (Arica), 39(3), 83-94. http://dx.doi.org/10.4067/S0718-34292021000300083

Olivares B, Vega A, Calderón MAR, Rey JC, Lobo D, Gómez JA, Landa BB. 2022. Identification of Soil Properties Associated with the Incidence of Banana Wilt Using Supervised Methods. Plants, 11(15):2070. https://doi.org/10.3390/plants11152070

Olivares, B., Hernández, R. 2020. Application of multivariate techniques in the agricultural land’s aptitude in Carabobo, Venezuela. Tropical and Subtropical Agroecosystems, 23(2):1-12. http://dx.doi.org/10.56369/tsaes.3233

Olivares, B., Pitti, J., Montenegro, E. 2020. Socioeconomic characterization of Bocas del Toro in Panama: an application of multivariate techniques. Revista Brasileira d

---

## [Editor Report · Decision Letter 1]

17 Sep 2024

A socioeconomic and cost benefit analysis of Tropical Race 4 (TR4) prevention methods among banana producers in Colombia

PONE-D-24-08984R1

Dear Dr.Thea Ritter,

We’re pleased to inform you that your manuscript has been judged scientifically suitable for publication and will be formally accepted for publication once it meets all outstanding technical requirements.

Kind regards,

Rajappa Janyanaik Joga, PhD

Academic Editor

PLOS ONE
---

## [Editor Report · Acceptance letter]

27 Sep 2024

PONE-D-24-08984R1 

PLOS ONE

Dear Dr. Ritter, 

I'm pleased to inform you that your manuscript has been deemed suitable for publication in PLOS ONE. Congratulations! Your manuscript is now being handed over to our production team.

Kind regards, 

on behalf of

Dr. Rajappa Janyanaik Joga 

Academic Editor

PLOS ONE